# A resource-efficient method for repeated HPO and NAS

**Giovanni Zappella**                                    ZAPPELLA@AMAZON.DE
*Amazon Web Services, Berlin, Germany*

**David Salinas**                                        DSALINA@AMAZON.FR
*Amazon Web Services, Berlin, Germany*

**Cedric Archambeau**                                    CEDRICA@AMAZON.DE
*Amazon Web Services, Berlin, Germany*

## Abstract

We consider the problem of repeated hyperparameter and neural architecture search (HNAS).We propose an extension of Successive Halving that leverages information gained in previous HNAS problems with the goal of saving computational resources. We empirically demonstrate that our solution is robust to negative transfer and drastically decreases cost while maintaining accuracy. Our method is significantly simpler than competing transfer learning approaches, setting a new baseline for transfer learning in HNAS.

## 1. Introduction

When solving a prediction problem with deep learning it is important to search over network architectures and to tune the network hyperparameters. However, conditions under which a deep neural network is deployed might drift away from the experimental conditions considered when it was developed. Hence, best practice recommends periodically updating the network architecture and retuning its hyperparameters. Unfortunately, practitioners often do not perform hyperparameter and architecture search (HNAS) once they have settled on a tuned deep neural network, because the process of updating and retuning the network can be time-consuming and costly (e.g., see Klein and Hutter, 2019, Fig.5). Several research groups proposed solutions for transfer learning in HNAS (see related work in Appendix A) to ease this process, but most assume a large number of historical evaluations are available. In many real-world scenarios, sharing evaluations can be problematic as the selected architectures and hyperparameter configurations can capture confidential information, such as click-rates or churn-rates, and historical evaluations might reside in models built by competing industrial players. In this work, we propose an extension of Successive Halving (Jamieson and Talwalkar, 2016; Karnin et al., 2013) which can reduce the usage of computational resources when a sequence of HNAS problems have to be solved. The method combines the robustness of testing a large set of candidate configurations and an aggressive pruning strategy which leverages the best evaluations from previous tuning jobs. This combination allows the algorithm to work with a small amount of previous evaluations in a reliable way. Our experiments show that our approach provides great savings when historical evaluations are available, regardless of their amount, and is robust to negative transfer (Wang et al., 2019).

## 2. Problem setup

We frame repeated HNAS as a sequence of Best Arm Identification (BAI) problems (see Audibert and Bubeck, 2010; Lattimore and Szepesvári, 2020, Ch. 33, and references therein) where the total number of steps in the sequence is unknown. Each of the BAI problems follow the non-stochastic setting described by Jamieson and Talwalkar (2016). At a high level, each combination of hyperparameters corresponds to an arm in the bandit problem and each arm pull can be seen as the amount of resources consumed (e.g., one epoch) for training the model with these hyperparameters. The arms are selected by an independent procedure (e.g., uniform random sampling) and provided to the tuning algorithm, which is agnostic to this external procedure. The setting used to define these problems is the non-stochastic bandit setting: the losses suffered by the learner are not sampled from an unknown fixed distribution as in the stochastic bandit setting, but become smaller over time and thus subject to a less restrictive assumption. For example, if a set of arms is defined as the number of neurons per layer and the number of layers of a neural network, we will train several neural networks in parallel using different combinations of these values. Each network will be trained for a number of epochs which is the number of pulls assigned to the arm representing it. In this example it is easy to see that the initial validation loss of the network will be large and will get smaller the more we train the network, until convergence is reached (see $\nu$ below).

More formally, in the non-stochastic BAI problem, the learner is provided with a set of arms $A = \{a_1, \ldots, a_k\}$. If the learner decides to play arm $a$ at time $t$, it will suffer a loss $\ell_{a,t} \in \mathbb{R}$. We assume that for each arm $a$, there exists $\nu_a$ such that $\nu_a = \lim_{t \to +\infty} \ell_{a,t}$. The goal of the learner is to identify $\arg\min_{a \in A} \nu_a$ by using at most a pre-defined number of arm pulls. The existence of the limit for the losses implies the existence of a non-increasing function $\gamma_a$ that bounds the distance to the limit with $|\ell_{a,t} - \nu_a| \leq \gamma_a(t)$. We define $\gamma_a^{-1}(\alpha) = \min\{t \in \mathbb{N} : \gamma_a(t) \leq \alpha\}$ which gives the smallest $t$ required to reach a given distance to the limit and $\bar{\gamma}^{-1}(\alpha) = \max_{a=1,\ldots,k} \gamma_a^{-1}(\alpha)$. Without loss of generality, we assume $\nu_1 = \min_{i \in A} \nu_i$ and define $\Delta_a = \nu_a - \nu_1$, as well as $\tau_a = \min_{\tau \in \mathbb{N}^+} s.t. \gamma_a(\tau) + \Delta_a > \gamma_1(\tau)$. Our setting extends the non-stochastic BAI problem by considering a sequence $1, \ldots, S$ of such problems. Arms identifiers remain unique across problems but no assumptions are made and their performance can vary over the sequence. Let $\tilde{a}_s$ be the arm identified as the best one by the learner. The goal is to minimize $\sum_{s=1}^{S} \Delta_{\tilde{a}_s}$ without exceeding the budget $B$ of arm pulls allocated for each problem.

## 3. Method

In order to provide a simple and effective algorithm for the repeated HNAS problem we make some additional assumptions. Winkelmolen et al. (2020) empirically demonstrate that a small set of hyperparameters configurations can provide effective solutions for an heterogeneous set of tuning tasks. Along the same lines, we assume that the size of the set of optimal configurations $A_S^*$ for a sequence of $S$ tasks satisfies $|A_S^*| < S$. The purpose of this assumption is to have an algorithm using a fixed maximum amount of resources over all the tasks in the sequence (otherwise the resources consumption would grow logarithmically in total number of arms). In practice, depending on the nature of the application, several heuristics can be

used to control the size of $A_S^*$ when sequences are long. For instance, one could use random subsampling, a FIFO queue, or cluster the configurations. We will further assume that configurations previously identified as optimal will quickly be either identified as optimal or outperformed by the optimal configuration. Hence, define $z$ s.t. $\forall a \in A_S^*, \tau_a \leq z$. This assumption formalizes the empirical observations where configurations identified as optimal on a task can perform very poorly on another one, quickly becoming inferior to the best configuration on the current task. When that is not the case, they are often optimal or very close to optimal. For convenience we define $n$ to be the cardinality of the union between the arms provided for the problem at hand and the arms identified as optimal in the previous steps.

Based on those assumptions we can propose a variant of Successive Halving (SH) (Jamieson and Talwalkar, 2016; Karnin et al., 2013), called Repeated Unequal Successive Halving (RUSH), using the previously discovered optimal arms in $A_S^*$ to determine if the optimal solution for the problem at hand has already been observed or not. The algorithm, formally described in Algorithm 1, works as follows: for each BAI problem, a set of configurations (arms) is sampled uniformly at random from the search space. RUSH splits the budget in equal parts to be consumed sequentially. At each step in the sequence the available budget is allocated uniformly on the available arms. At the end of the step the number of candidates is reduced by a factor $\eta$ according to their performance. Moreover, RUSH stops the exploration of any arm under-performing an arm in $A_S^*$, saving resources. This possible because the value of $\tau$ for every arm in $A_S^*$ is buonded by $z$ and the algorithm receives a budget as described in Theorem 1. We can also show that RUSH is able to correctly identify the best arm for each problem in the sequence. The proof is available in Appendix B.

---

**Algorithm 1** Repeated Unequal Successive Halving (RUSH)

---

**Input:** $\eta$ (halving hyper-parameter), $B$ (budget)
$|A_0^*| \leftarrow \emptyset$
$s \leftarrow 0$
**while** a new task is available **do**
    $A_s^{new} \leftarrow$ set of new arms
    $A_s^0 \leftarrow A_s^{new} \cup A_s^*$
    $n \leftarrow |A_s^0|$
    **for** $k = 0, \ldots, \lceil \log_\eta n \rceil - 1$ **do**
        $\forall a \in A_s^k$, pull it $\left\lfloor \frac{B}{\max(1, \lfloor n/\eta^{k+1} \rfloor) \lceil \log_\eta n \rceil} \right\rfloor$ times
        $\forall a, r_a \leftarrow$ position of $a$ in ranking by loss
        $r^* \leftarrow \min(r_i), \forall i \in A_s^*$
        $A_s^{k+1} =$
        $\{i \in A_s^k : r_i < \max(\min(r^* + 1, \lfloor n/\eta^{k+1} \rfloor), 1)\}$
    **end for**
    $\tilde{a} \leftarrow$ best arm from $A_s^{\lceil \log_\eta n \rceil - 1}$
    **if** $\tilde{a} \notin A_s^*$ **then**
        $A_{s+1}^* \leftarrow A_s^* \cup \{\tilde{a}\}$
    **end if**
    $s \leftarrow s + 1$
**end while**

---

**Theorem 1.** *If the budget $B$ provided to the algorithm for each step of the sequence $1, \ldots, S$ is larger than $\lceil \log n \rceil \max \left( 2n + \sum_{a=2,\ldots,n} \bar{\gamma}^{-1}(\Delta_a/2), zn \right)$ then Algorithm 1 will correctly identify the best arm.*

It is possible to combine RUSH with non-uniform sampling (e.g., Klein et al., 2020), or other transfer learning techniques (e.g., Perrone et al., 2019) since RUSH and SH are agnostic to the process generating the set of candidate arms.

## 4. Experiments

In order to perform experiments in a realistic environment we created sequences of tuning tasks obtained by training the same algorithms on different, but related, datasets. We created six collections of datasets: 3 obtained by pre-processing the same dataset in 20 different ways and tuning the hyperparameters of an XGBoost classifier, 2 obtained from public NAS benchmarks and 1 obtained mixing tasks from the three XGBoost-based collections described above in order to test the robustness of the approach. All the details about the creation of the datasets are available in Appendix C. For every experiment, we created sequences of 20 tuning tasks training the same algorithm on a random permutations of the datasets in the considered collection (without replacement if the collection is large enough). We repeated this operation 25 times for a total of 500 tuning jobs for each experiment. Results were averaged both over the 25 runs and the 20 datasets in each collection, obtaining a single predictive performance and cost for each pair (tuning algorithm, dataset collection). We compare RUSH and a version of Hyperband using RUSH as subroutine instead of SH (called HB-RUSH) with the ones obtained by state-of-the-art solutions: **Successive Halving (SH)** (Jamieson and Talwalkar, 2016), from which we derived RUSH, a bandit algorithm for non-stochastic best arm identification; **Hyperband (HB)** (Li et al., 2016), a bandit method for hyper-parameter tuning using Successive Halving as subroutine with different level of resources allocation; and we also compared our approach with a different class of algorithms, the **Bounding Box (BB)** approach by Perrone et al. (2019). The BB approach is not designed for the same setting of RUSH, so we provide for every tuning job in the sequence all the information about all the remaining tuning jobs (past and future), moreover we do not provide the results of the evaluations obtained with the optimizer but the "ground truth" values of all the configurations evaluated. This makes the comparison unfair, since RUSH will have access to information from the same number of tuning tasks only in the last step of the sequence and it will never have certainty about the performance of the optimal configuration. On the other hand, even if the comparison is unfair and is not always possible to run BB in practice, it will provide a good reference point. We selected the BB approach because it is the best performing algorithms in the Hyperband experiments in (Perrone et al., 2019). BB is not used in the experiment in Appendix 4.3 since it does not provide any protection against negative transfer. In our comparison, we set the hyper-parameter $\eta = 3$ for all the instances of RUSH, SH and HB. The budget for HB-RUSH and HB is set as described in (Li et al., 2016) and for RUSH and SH it is equivalent to a single bracket of HB.

### 4.1 Predictive performance comparison.

The results in Table 1 show that the performance of RUSH are on par with the performance of SH, with differences in the order of the standard deviation on most datasets and tiny advantages for one or the other in the other cases. A similar trend can be observed for HB-RUSH and HB. The standard deviation values are reported in Appendix E and, while some variance is present, it does not seem to prevent the usage of this method in real-world applications. Autorange(BB) displays a trend of slightly better results, but for most datasets the differences are in the order of the standard deviation. It is important to observe that RUSH performance are on-par also on BALMIX, which is an heterogeneous set of tasks.

| Dataset | BALMIX | BANK | CCDEFAULT | COVTYPE | FCNET | NAS201 |
|---|---|---|---|---|---|---|
| SH-Autorange(BB) | 0.630 | 0.691 | 0.950 | 0.181 | 0.077 | – |
| HB-Autorange(BB) | 0.611 | 0.666 | 0.932 | 0.174 | 0.075 | – |
| RUSH | 0.546 | 0.693 | 0.978 | 0.209 | 0.079 | 31.897 |
| SH | 0.553 | 0.698 | 0.980 | 0.210 | 0.080 | 32.433 |
| RUSH-HB | 0.537 | 0.678 | 0.968 | 0.186 | 0.075 | 31.355 |
| HB | 0.542 | 0.691 | 0.978 | 0.186 | 0.078 | 31.908 |

Table 1: Average prediction performance obtained by different optimizers. For BANK and CCDEFAULT we report 1-F1, for COVTYPE we report 1-F1micro, for FCNET and NAS201 the prediction error. In all cases lower is better.

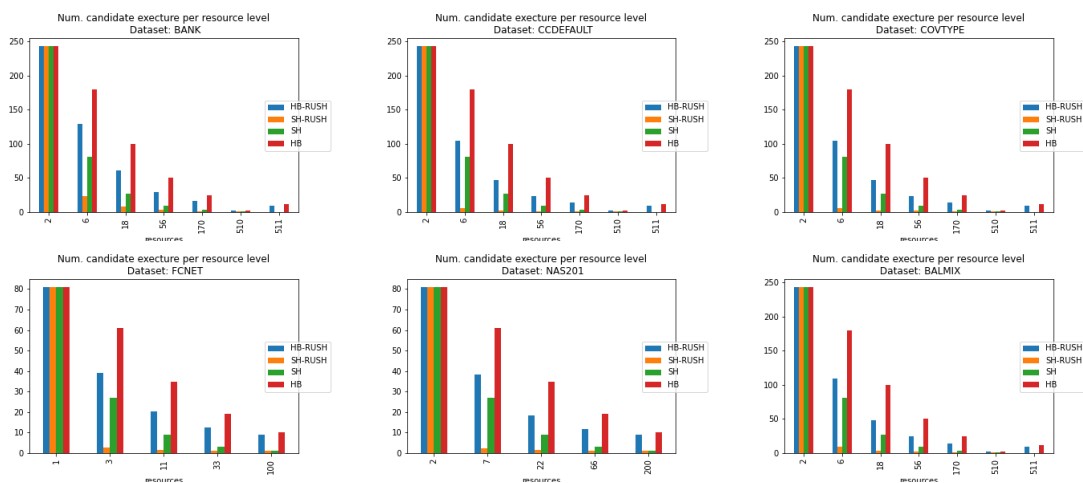

Figure 1: Number of evaluated candidates per resources level. Lower is better.

Results showing the robustness of RUSH performances to the choice of the budget are available in Appendix G.

## 4.2 Resources consumption comparison.

After observing that the predictive performance of the models produced with RUSH are on-par with other optimizers, we studied the resources consumption of different optimizers. To this purpose, we track the number of configurations evaluated by different optimizers for different resources levels (e.g., how many configurations were evaluated for a certain number of epochs). The number of configurations evaluated is a proxy metric for the consumption of computational

| Dataset | RUSH vs SH | HB-RUSH vs HB |
|---|---|---|
| BALMIX | 41.044 | 18.336 |
| BANK | 34.465 | 17.345 |
| CCDEFAULT | 45.627 | 18.646 |
| COVTYPE | 45.932 | 26.941 |
| FCNET | 0.587 | 0.657 |
| NAS201 | 48.547 | 19.357 |

Table 2: Average time reduction (in percentage) achieved by RUSH and HB-RUSH compared to SH and HB. Higher is better.

resources (more configurations evaluated, more resources used). The results reported in the Figure 1 show a very clear trend: HB-RUSH and RUSH require less evaluations than SH and HB, in some cases even an order of magnitude less. Since in modern cloud services

customers get billed according to the time spent using the computational resources, and since lower computational resources usage could be a proxy for less resources used (e.g., energy), we would like to quantify the total computation time saved by using RUSH. The results in Table 2 mostly confirm the findings reported in the first part of this section with massive reduction of consumed resources on most datasets. There is a notable exception on which RUSH provides only a small advantage: FCNET. This is due to the huge variance in the time-per-epoch observed for the configurations available for FCNET which can make extremely expensive to reach the first checkpoint of evaluation in the algorithm. An additional investigation is available in Appendix F.

### 4.3 Negative transfer experiment

To provide further evidence of the robustness of RUSH, we created an ad-hoc experiment with two tasks where the ranking by predictive performance is completely reversed. The best configuration on one task is the worst configuration on the other and vice versa. In this case the information provided by the previous tasks is not only useless but detrimental. Considered one of the tasks in

| Dataset | BANK1 | INVBANK1 |
|---------|---------|----------|
| RUSH | 0.610755 | 0.0 |
| SH | 0.609984 | 0.0 |
| HB-RUSH | 0.604140 | 0.0 |
| HB | 0.604886 | 0.0 |

Table 3: Predictive performance of the models produced by HPO with different optimizers and tasks.

the BANK dataset collection, we created the inverse (called INVBANK1) simply by setting the F1 score of evaluation of each configuration to 1-F1. This is a worst-case scenario which will prevent many transfer learning algorithms from identifying the optimal solution (e.g., Autorange) or will cause a significant cost increase since the optimizer will start evaluating poorly performing configuration (e.g., BO-related methods). The sequences in this case are composed only by the tasks BANK1 and INVBANK1 and the order is randomly selected. Results are averaged on 25 repetitions. The results reported in Table 3 show that, in fact, RUSH and HB-RUSH match the performance of their "standard" counterparts both in terms of predictive performance. Also, this performance is achieved without additional cost, see results in Appendix H.

### 5. Conclusion and future work

In this work we introduced a variant of Successive Halving targeting the common use case of repeated HNAS. We characterized the problem from a formal perspective and provided an effective algorithm for solving it. We tested the new algorithm, RUSH, on a number of different benchmarks involving both Neural Networks and tree-based predictors. RUSH reduced the usage of compute required for HNAS. In most cases we observed a double digit cost reduction and on-par predictive performance. We also created mixed tuning jobs collections (i.e., BALMIX) and ad-hoc experiments to demonstrate empirically the robustness of RUSH to negative transfer. We observed lesser improvements when the variance of the evaluation time of the configurations became extremely high, suggesting that a cost-sensitive version of RUSH following the one already created for SH and HB (Ivkin et al., 2021) could be of interest. Another research direction would be to consider RUSH-like approaches for HNAS in the continual learning setting (e.g., see Aljundi et al. (2019); Borsos et al. (2020)).

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

## Appendix

## Appendix A. Related work

In the AutoML literature there are few important families of methods doing transfer learning for HPO and NAS problems. In the following we will present the main approaches:

- **Transfer learning for Bayesian Optimization (BO).** There are a number of approaches for transfer learning in Bayesian Optimization (e.g, Joy et al. (2019); Feurer et al. (2015); Perrone et al. (2018); Salinas et al. (2019)) but they are often focusing on scenarios with large amounts of evaluations from which to transfer (e.g., see Feurer et al. (2015); Perrone et al. (2018)) or do not provide guarantees about the resources consumption to achieve "optimal" performance in presence of negative transfer (e.g., Salinas et al. (2019)). Algorithms such Successive Halving or Hyperband are significantly more resource-efficient (Falkner et al., 2018), also because they make a different trade-off between end to end wall-clock time and resources consumption. This trade-off favors a large number of parallel computations and shorter waiting time, which lead to the usage these algorithms in different practical scenarios, making their comparison of limited usefulness from a practical perspective.

- **Transfer learning for Hyperband.** Transfer learning for Hyperband-style algorithms is not an extensively studied topic. Methods such the one in (Valkov et al., 2018) strongly focus on increasing predictive performance instead of decreasing the consumption of resources.

- **Learning the search space** (Perrone et al., 2019) is an effective way to sample configurations leading to good results, especially when the search space is not correctly specified. On the other hand, this approach can have some difficulties when the number of previous HPO evaluations is extremely small or when the set of HPO evaluations are performed on an heterogeneous set of tasks. Moreover, this kind of improvements can be combined with a large number of HNAS optimizers, including the one presented in this paper.

The work described in (Stoll et al., 2020) on the surface may look similar to the setting considered in this work but there are substantial differences. The main differences is that we consider a sequence of tasks as an "adjustments" to data transformation procedures, pre-processing pipelines, etc., while the HT-AA setting is mostly focusing on changes to hardware, ML algorithms and search space (Section 2). While a changing search space presents an interesting scientific problem and our approach can be adapted to that, a fixed search space for a sequence of tasks is a perfectly valid assumption for applications we are interested in. Moreover, our evaluation is not designed to reach a pre-defined level of accuracy, but to reach the same performance of Successive Halving using a reduced amount of resources. A detail that contrast with (Stoll et al., 2020) (Section 5), where the authors report a trend of decreasing speedup when the target objective becomes "more optimal". Last but not least, the HT-AA setting is currently considered for transfer from a single task to another, while in our case there is no previous knowledge about the similarity among the tasks and the sequences of tasks presented to the learner in our experiments can be extremely heterogeneous.

## Appendix B. Proof of Theorem 1

**Theorem.** *If the budget $B$ provided to the algorithm for each step of the sequence $1, \ldots, S$ is larger than $\lceil \log n \rceil \max \left( 2n + \sum_{a=2,\ldots,n} \bar{\gamma}^{-1}(\Delta_a/2), zn \right)$ then Algorithm 1 will correctly identify the best arm.*

*Proof.* This results follows from the fact that when the first term in the max function is larger, then we are exactly in the same case of (Jamieson and Talwalkar, 2016, Theorem 1). When the second argument is larger, then we are guaranteed that no arm will be eliminated before each arm (including the ones in $|A^*|$) has been pulled at least $z$ times and so, by definition of $z$, there is no chance for to discard the optimal arm by eliminating the arms performing worse than the ones in $|A^*|$ and since the budget is larger than what is required by Successive Halving, there is no risk in discarding the arms at the bottom of the ranking. $\square$

## Appendix C. Datasets

For the purpose of these experiments we used two different families of datasets. The first family comprises public benchmarks developed for the NAS problem:

- **FCNET**: a collection of NAS problems where the learner needs to identify the best architecture for a simple neural network on four different datasets. The details about this collection of benchmarks are provided in (Klein and Hutter, 2019).

- **NAS201**: a collection of NAS problems where the learner needs to identify the best architecture for a neural network on three different computer vision datasets (CIFAR10-valid, CIFAR100 and IMAGENET). The details about this collection of benchmarks are provided in (Dong and Yang, 2020).

These datasets are not specifically designed to benchmark algorithms in our setting but they are useful to our purposes since they are widely know and provide a clear reference point to better understand our results.

The second family is a set of benchmarks developed starting from public datasets and applying different pre-processing steps as a data scientists would do. The selected datasets, all from UCI are the following:

- **CCDEFAULT** (Yeh and Lien, 2009): a binary classification dataset for which the learner has to predict the default of a credit card customer. The features of each customers involve demographic information and payments-related information, both in numerical and categorical form.
  `https://archive.ics.uci.edu/ml/datasets/default+of+credit+card+clients`

- **BANK** (Moro et al., 2014): a binary classification dataset from which the learner has to predict responses to marketing campaigns sent by a Portuguese banking institution. Features are both categorical and numerical. The feature causing target leakage has been dropped.
  `https://archive.ics.uci.edu/ml/datasets/Bank+Marketing`

- **COVTYPE**: a multi-class classification dataset for which the learner has to predict the vegetation type given information regarding the piece of land considered (e.g., altitude). The data points only have numerical features and are categorized in seven classes.
https://archive.ics.uci.edu/ml/datasets/covertype

For each one of these public datasets we created a collection of 20 derived datasets applying different pre-processing techniques to different attribute columns. The pre-processing methods applied to the different features were selected independently and uniformly at random according to the kind of feature considered (categorical or numerical). We only used preprocessors acting on a single attribute/feature but nothing prevents the usage of our approach with more complex preprocessing pipelines.

Preprocessing options available for categorical attributes:

- **ONE HOT ENCODER**: encodes categorical features using one binary feature per category.

- **BACKWARD DIFFERENCE ENCODER**: contrast coding of categorical variables. Features with zero variance, if taken into consideration, are dropped.

- **ORDINAL ENCODER**: encodes categorical features into one single ordered feature.

- **BASE-N ENCODER**: encodes the categories into arrays of their base-N representation.

- **DROP**: eliminates the considered feature.

Preprocessing options available for numerical attributes:

- **STANDARD SCALER**: transforms the features subtracting the mean and dividing by the standard deviation.

- **MINMAX SCALER**: transforms the features by subtracting the smallest value and dividing by the size of the (min,max) range.

- **BINARIZER**: transforms values to zero if the are below a certain threshold, to one otherwise. The mean value is used as a threshold.

- **QUANTILE TRANSFORMER**: transforms numerical features by replacing them with their quantile identifier.

- **DROP**: eliminates the considered feature.

The implementations of the categorical features transformers are available in the Category Encoders package[1] and the ones for the numerical features transformer are available in Scikit-learn (Pedregosa et al., 2011). When not specified, the default hyper-parameters are used.

To summarize, for each attribute of the original dataset, the entire column is selected,

---

1. https://contrib.scikit-learn.org/category_encoders/

according to the type of the data, we sample a feature transformer uniformly at random from a list of five, the transformation is applied, the output of the transformer is added to the dataset and the original values fed to the transformer are dropped. This process is repeated 20 times for each original dataset, generating 20 different datasets from each of the UCI datasets listed above. The BALMIX collection is obtained sampling datasets from the collections derived from BANK, CCDEFAULT and COVTYPE. For each of the 20 datasets, we trained XGBoost on 1000 randomly sampled configurations (the same for all the datasets) using the number of rounds as resources level, tracking the predictive performance using 1-F1 (or 1-F1micro for multi-class) for 512 rounds.

The configurations sampled by the optimizers can sampled uniformly at random from the search space and the results of their evaluation is obtained by interpolation with nearest neighbor.

## Appendix D. XGBoost search spaces

The following search space has been used in our experiments with XGBoost.

Hyperparameters:

- hp_max_depth: integer in $[1, 8]$

- hp_learning_rate: float in $[0.001, 1.0]$ log-scale

- hp_reg_lambda: float in $[0.000001, 2.0]$ log-scale

- hp_gamma: float in $[0.000001, 64.0]$ log-scale

- hp_reg_alpha: float in $[0.000001, 2.0]$ log-scale

- hp_min_child_weight: float in $[0.000001, 32.0]$ log-scale

- hp_subsample: float in $[0.2, 1.0]$

- hp_colsample_bytree: float in $[0.2, 1.0]$

- hp_tree_method: categorical in $["exact", "hist", "approx"]$

## Appendix E. Standard deviation for predictive performance

In this section we report the standard deviation for the predictive performance measurements reported in Table 1.

## Appendix F. Deep dive on FCNET

We further investigate the results obtained by SH and RUSH on FCNET. We identified a number of hyperparameters configurations with a disproportionately high cost being sampled by both SH and RUSH (see Table 5). Moreover, these extremely costly configurations often do not provide particularly good performance and get pruned at the end of the first rung. This leads to much cheaper rungs after the first one, and the overall time-cost being dominated by the first rung. While we have are not in the position of investigate this further,

| Optimizer | Dataset | Stdev |
|---|---|---|
| Autorange(BB)+SH | BALMIX | 0.254994 |
| | BANK | 0.022924 |
| | CCDEFAULT | 0.030943 |
| | COVTYPE | 0.044704 |
| | FCNET | 0.000146 |
| Autorange(BB)+HB | BALMIX | 0.246499 |
| | BANK | 0.012394 |
| | CCDEFAULT | 0.023578 |
| | COVTYPE | 0.046303 |
| | FCNET | 0.000112 |
| RUSH | BALMIX | 0.353384 |
| | BANK | 0.024797 |
| | CCDEFAULT | 0.010956 |
| | COVTYPE | 0.068631 |
| | FCNET | 0.000101 |
| | NAS201 | 0.263551 |
| SH | BALMIX | 0.349033 |
| | BANK | 0.021733 |
| | CCDEFAULT | 0.008713 |
| | COVTYPE | 0.065947 |
| | FCNET | 0.000124 |
| | NAS201 | 0.626832 |
| HB-RUSH | BALMIX | 0.349084 |
| | BANK | 0.035077 |
| | CCDEFAULT | 0.023711 |
| | COVTYPE | 0.064896 |
| | FCNET | 0.000043 |
| | NAS201 | 0.184094 |
| HB | BALMIX | 0.356893 |
| | BANK | 0.025858 |
| | CCDEFAULT | 0.010542 |
| | COVTYPE | 0.070216 |
| | FCNET | 0.000043 |
| | NAS201 | 0.282075 |

Table 4: Standard deviation on the average performance obtained in the first experiment

it is possible that the huge time variance could be due to external events happening during the training process (e.g., other processes running on the same machine). These extreme conditions are not present in NAS201. Since none of the algorithm used in our experiments directly controls the time-cost associated with the evaluation of different configurations, but RUSH relied to lower the numbers of configurations evaluated to reduce cost, the possibilities

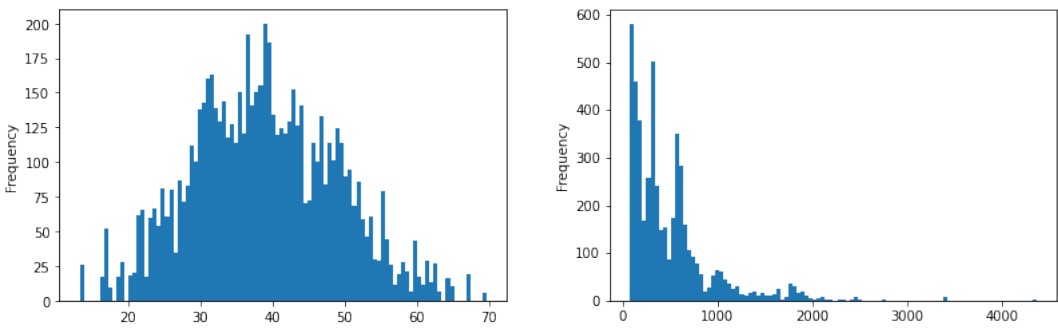

Table 5: Distribution of the cost of different configurations at the minimum resources level.

to directly impact the total cost are limited. An interesting scientific challenge will be to design an algorithm explicitly controlling the cost.

## Appendix G. Results with different budget values

To validate the robustness of RUSH respect to the available budget, we run a comparison between RUSH and SH with different budget levels. The results reported in 6 show that in some cases RUSH can be slightly better but, in general, there is no significant difference between RUSH and SH.

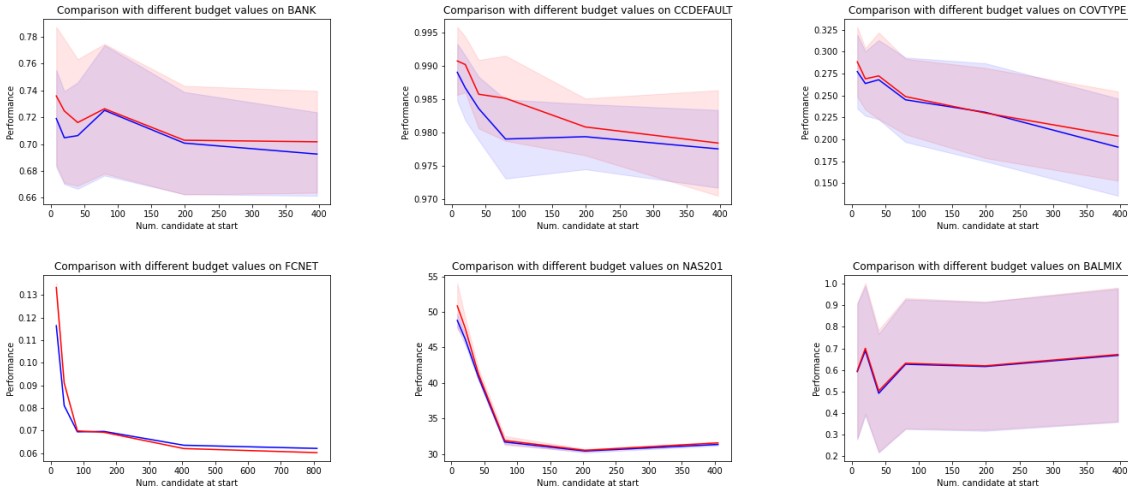

Table 6: Performance of the model vs budget allocated for the optimizer. Each point on the x-axis correspond to a different budget provided to the optimizer and each point on the y-axis correspond to the best performance obtained when the budget of the optimizer is exhausted. Results are averaged on all the datasets of the same category.

## Appendix H. Negative transfer additional results

In this section we report the time difference between RUSH, Successive Halving and Hyperband. Higher percentages means significant cost reductions, while percentages close to zero mean there is no significant difference.

| Dataset | Time SH vs SH-RUSH | Time HB vs HB-RUSH |
|---|---|---|
| BANK | -0.0609 | -0.5168 |
| INVBANK | -0.2811 | -1.0856 |

Table 7: Time gain, in percentage, observed by comparing SH with RUSH and HB with HB-RUSH.

