# OpenReview forum: "A resource-efficient method for repeated HPO and NAS problems"
_ICML.cc/2021/Workshop/AutoML — AutoML@ICML2021 Poster_

### Official Review · Reviewer_Twcp · 2021-06-11
**A resource-efficient method for repeated HPO and NAS**

**Rating:** 6
**Confidence:** 4

**Review:**

The paper describes an approach to solving the "repeated Best Arm Identification" problem by maintaining and updating a portfolio of candidate configurations from previous runs. The problem setting described is somewhat intuitive and might be relevant in practice.
The proposed solution is simple and seems to obtain good performances in comparison to
the employed baselines. It only requires a minor modification to the SH / HB procedure in order to maintain and update the portfolio and additionally employs more aggressive pruning based on the performance of portfolio configurations.


Major Comments
  - The employed pruning strategy seems rather aggressive, potentially killing off all but the best configuration in the portfolio.
    Could less aggressive schedules lead to more stable behaviour?

  - The notation in the algorithm is somewhat scrambled up is r(i) a function or the rank?
    I think what r* should say is the minimum rank of all action in A*.
    The current notation suggests r(i) = 1 always (since you take the ranks only over A*)

  - 4.1: You state that you ran 25 replications, why not report standard errors directly and only do this in the appendix? This is a crucial detail for the evaluation of the methodology. It can furthermore provide viable information especially when differences are small.
    Reported standard errors from the appendix are comparatively large,  so the
    big questions remain is how reliable the method is in practice?
  I think that should be discussed in the paper. since it could be a major limitation.

  - I can imagine that performance drastically depends on the order of tasks, is this what induces the variance in performance in your experiments?

  - I think the negative transfer learning scenario is interesting, however, I do not understand the results. Is there an error in Table 4? What is 0.0 for INVBANK  supposed to tell me?
    Furthermore, I feel the scenario considered is a little too simplistic because the size
    of your portfolio will always be 1?

  - I agree that the cited work (Stoll et al., 2020) is slightly different in scope, but not to the degree the authors give it credit for. Your example changes in data due to differing pre-processing pipelines. Considering the joint problem of configuring pipeline and ML algorithm hyperparameters,
    changes to the pipeline (as in your experiments) can correspond to changes
    in the full pipeline's hyperparameter space.
    I would therefore consider adding the baselines from this work as additional
    points of comparison

  - Maybe I am missing something, but the statement about z 1) does seem tautological and
    2) it is not intuitive to me, why this should convey what the following sentences
    suggest it does.

  Typos & Minor Comments

  - Page 2: Typo: the the smallest t
  - Page 2: gamma_1(t) is not defined, I assume it is the gamma_a(t), such that nu(a) = nu1?
  - Page 3: Typo: on a tasks; buonded by z;
  - Table 2: This is a Figure not a table? What is an executure? x-labs are cut off?

---

### Official Review · Reviewer_L7gV · 2021-06-17
**A resource-efficient method for repeated HPO and NAS**

**Rating:** 7
**Confidence:** 5

**Review:**

This paper introduces an extension of Successive Halving (RUSH) targeting the common use case of repeated Hyperparameter tuning and architectural search. It leverages information gained in previous HNAS problems with the goal of saving computational resources.

Authors show that RUSH reduced the usage of compute required for HNAS.

Authors show a comparison with other similar algorithms and results are positive. There is a reduction of resources needed while either maintaining or improving the quality of the model.

It'd be interesting to get the standard deviation or average error for the multiple runs to get an idea of how overlapping results are.

---

### Decision · Program_Chairs · 2021-06-21

Accept (Poster)